# High Incidence of Inappropriate Alarms in Patients with Wearable Cardioverter-Defibrillators: Findings from the Swiss WCD Registry

**DOI:** 10.3390/jcm10173811

**Published:** 2021-08-25

**Authors:** Boldizsar Kovacs, Haran Burri, Andres Buehler, Sven Reek, Christian Sticherling, Beat Schaer, Andre Linka, Peter Ammann, Andreas S. Müller, Omer Dzemali, Richard Kobza, Matthias Schindler, Laurent Haegeli, Kurt Mayer, Urs Eriksson, Claudia Herrera-Siklody, Tobias Reichlin, Jan Steffel, Ardan M. Saguner, Firat Duru

**Affiliations:** 1Department of Cardiology, University Heart Center Zurich, Rämistrasse 100, 8091 Zurich, Switzerland; andres.buehler@usz.ch (A.B.); matthias.schindler@usz.ch (M.S.); jan.steffel@usz.ch (J.S.); ardan.saguner@usz.ch (A.M.S.); firat.duru@usz.ch (F.D.); 2Department of Cardiology, University Hospital of Geneva, 1205 Geneva, Switzerland; Haran.Burri@hcuge.ch; 3Hirslanden Klinik Aarau, 5001 Aarau, Switzerland; Sven.Reek@hirslanden.ch; 4Department of Cardiology, University Hospital Basel, 4031 Basel, Switzerland; Christian.Sticherling@usb.ch (C.S.); beat.schaer@usb.ch (B.S.); 5Department of Cardiology, Cantonal Hospital Winterthur, 8400 Winterthur, Switzerland; andre.linka@ksw.ch; 6Department of Cardiology, Cantonal Hospital St. Gallen, 9007 St. Gallen, Switzerland; peter.ammann@kssg.ch; 7Department of Cardiology, Triemli Hospital Zurich, 8063 Zurich, Switzerland; AndreasStephan.Mueller@triemli.zuerich.ch; 8Department of Cardiac Surgery, Triemli Hospital Zurich, 8063 Zurich, Switzerland; Omer.Dzemali@triemli.zuerich.ch; 9Department of Cardiology, Cantonal Hospital Lucerne, 6004 Lucerne, Switzerland; richard.kobza@luks.ch; 10Department of Cardiology, Cantonal Hospital Aarau, 5001 Aarau, Switzerland; laurent.haegeli@ksa.ch; 11Department of Cardiology, Cantonal Hospital Graubünden, 7000 Chur, Switzerland; Kurt.Mayer@ksgr.ch; 12Department of Cardiology, GZO Regional Healthcare Center Wetzikon, 8620 Wetzikon, Switzerland; urs.eriksson@gzo.ch; 13Department of Cardiology, University Hospital Lausanne, 1011 Lausanne, Switzerland; Claudia.Herrera-Siklody@chuv.ch; 14Department of Cardiology, Inselspital, University Hospital Bern, University of Bern, 3010 Bern, Switzerland; tobias.reichlin@insel.ch; 15Center for Integrative Human Physiology, University of Zurich, 8006 Zurich, Switzerland

**Keywords:** wearable cardioverter-defibrillator, alarm, inappropriate therapy, obesity, outcome

## Abstract

Background: The wearable cardioverter defibrillator (WCD) uses surface electrodes to detect arrhythmia before initiating a treatment sequence. However, it is also prone to inappropriate detection due to artefacts. Objective: The aim of this study is to assess the alarm burden in patients and its impact on clinical outcomes. Methods: Patients from the nationwide Swiss WCD Registry were included. Clinical characteristics and data were obtained from the WCDs. Arrhythmia recordings ≥30 s in length were analysed and categorized as VT/VF, atrial fibrillation (AF), supraventricular tachycardia (SVT) or artefact. Results: A total of 10653 device alarms were documented in 324 of 456 patients (71.1%) over a mean WCD wear-time of 2.0 ± 1.6 months. Episode duration was 30 s or more in 2996 alarms (28.2%). One hundred and eleven (3.7%) were VT/VF episodes. The remaining recordings were inappropriate detections (2736 (91%) due to artefacts; 117 (3.7%) AF; 48 (1.6%) SVT). Two-hundred and seven patients (45%) had three or more alarms per month. Obesity was significantly associated with three or more alarms per month (*p* = 0.01, 27.7% vs. 15.9%). High alarm burden was not associated with a lower average daily wear time (20.8 h vs. 20.7 h, *p* = 0.785) or a decreased implantable cardioverter defibrillator implantation rate after stopping WCD use (48% vs. 47.3%, *p* = 0.156). Conclusions: In patients using WCDs, alarms emitted by the device and impending inappropriate shocks were frequent and most commonly caused by artefacts. A high alarm burden was associated with obesity but did not lead to a decreased adherence.

## 1. Introduction

Patients with heart disease who are at high risk for sudden cardiac death (SCD) may be candidates for implantable cardioverter defibrillator (ICD) therapy. ICDs can treat ventricular tachycardia and ventricular fibrillation (VT/VF) by direct current cardioversion or defibrillation, if they are not amenable to overdrive pacing. While ICD therapies can be lifesaving, experiencing a shock while being conscious is often a painful and traumatizing event. Sophisticated device programming with various discrimination and therapeutic algorithms aim to reduce the number of shocks (appropriate or inappropriate), while maintaining the lifesaving capability of these devices [1,2,3,4].

The wearable cardioverter defibrillator (WCD) has been introduced for temporary protection against SCD for a limited time period in patients for whom ICD therapy is not indicated or possible. Although several registries have demonstrated that WCDs can prevent SCD [5,6,7,8,9,10], the VEST trial did not show a reduction in SCD and death from documented ventricular arrhythmia [11]. As a clear benefit from its use is currently lacking, possible harmful effects of WCD use necessitate further scrutiny.

The WCD uses surface electrodes built into the vest with direct skin contact to detect the underlying electrocardiogram of patients [12]. This approach is more prone to artefacts leading to inappropriate arrhythmia detection and shock delivery. A critical feature for the discrimination between artefacts, benign (supraventricular) arrhythmias and VT/VF are algorithms employed by the WCD.

Data on the incidence on alarms emitted by the WCD and their impact on patient outcomes and quality of life is lacking. The aim of this study is to fill this gap by studying the available data from the large Swiss WCD registry.

## 2. Methods

### 2.1. Swiss WCD Registry

Characteristics of the Swiss WCD Registry have been reported previously [6]. In short, it is a nationwide, retrospective registry collecting data from 12 participating hospitals of patients, who have a history of WCD use. In this study, we included WCD patients from December 2011 until February 2018.

### 2.2. WCD Arrhythmia Detection Sequence

Arrhythmia recognition by the WCD (LifeVest, ZOLL, Pittsburgh, Pennsylvania, USA) is based on a proprietary algorithm (TruVector^TM^) analysing the heart rate and QRS morphology [12]. Contrary to ICD, the WCD produces a series of warning signals prior to administration of a shock, and patients can press the response buttons to withhold shock application. The emitted alarms include a silent vibration alert, a loud siren alarm, and a voice command warning bystanders of impending shock administration. The duration of an alarm sequence is at least 30 s. A shock can be administered at earliest 30 s after arrhythmia recognition. Every arrhythmia detection leads to the same sequence of alarms (Figure 1). The nominal detection threshold of WCD is programmed as 150 beats per minute (bpm) for VT and 200 bpm for VF for sensitive detection of relevant arrhythmias. The WCD necessitates 5–6 s for arrhythmia recognition, followed by 10 s of arrhythmia confirmation. If VT/VF is confirmed, an alarm sequence is initiated until the patient presses the response button or a shock is administered. Therefore, any detected arrhythmic event leading to an alarm is at least 15 s in duration. All arrhythmic events leading to an alarm are stored on the proprietary LifeVest Network.

### 2.3. WCD Alarm Events

In this study, any arrhythmia detected by the WCD leading to an actual alarm sequence automatically was termed “alarm” (detected arrhythmia duration at least 15 s). Any arrhythmia detected by the WCD for more than 30 s was deemed potentially clinically relevant and was termed “recording”. All recordings were reviewed for underlying rhythm. Two investigators independently reviewed and adjudicated these recordings as (1) sustained VT (lasting 30 s or more on recording) or VF, (2) non-sustained VT (lasting less than 30 s on recording, (3) atrial fibrillation (AF), (4) supraventricular tachycardia, or (5) artefact. Disagreements were resolved by consensus or arbitration by a third investigator. Bradyarrhythmia events were infrequent and are not included in this analysis.

### 2.4. Statistical Analysis

A descriptive statistical analysis was performed on the available data set. Categorical variables were reported as frequencies (percentage), continuous variables as means (±standard deviation) or as medians (IQR, range). Statistical analysis was performed using chi square, Fisher’s exact test on categorical variables and univariate regression analysis on continuous variables, as appropriate. A two-sided *p* value < 0.05 was considered statistically significant. All statistical analyses were conducted using R 1.3.1073 (R Foundation for Statistical Computing, Vienna, Austria).

## 3. Results

The Swiss WCD Registry included data from 456 patients with a mean follow-up of 514 ± 384 days [6]. The mean effective wear-time of WCD (during which patients actually wore the WCD) was 2.0 ± 1.6 months. A total of 10653 alarms were registered during the observed period. Of these, 7657 alarms (71.9%) were triggered by arrhythmia detection shorter than 30 s and, accordingly, deemed clinically irrelevant. The length of detected arrhythmia episodes was 30 s or more in 2996 (28.1%) of these recordings. The mean alarm burden for the total population was 24.4 alarms per month per patient and the median alarm burden was 3 (IQR 0–17). One hundred and thirty-two patients (28.9%) had no alarms, and thus no recordings. Patients who had at least one alarm had a mean number of 34.3 alarms per month (Appendix A). Two-hundred and seven patients (45%) had three or more alarms per month. Baseline characteristics, including stratification of <3 vs. ≥3 alarms/month are shown in Table 1.

### 3.1. Analysed Recordings

Analysis of all recordings (*n* = 2996) revealed that 111 (3.7%, 1% of all alarms) were correctly identified as VT/VF episodes. These consisted of 89 sustained episodes (80.2%), 17 episodes with shock delivery (15.3%), and 5 non-sustained episodes (4.5%). Of note, the non-sustained episodes were likely detected, in part, due to artefact, as the actual arrhythmia duration was below 30 s. The remaining recordings were either artefacts (2736, 91%), AF (117, 3.7%) or supraventricular tachycardia (48, 1.6%) (Figure 2). There were no inappropriate shocks detected during the observed time period.

### 3.2. Variables Associated with High Alarm Burden

Patients with three or more alarms per month had a significantly higher BMI. This difference was mainly due to the higher number of patients with ≥3 alarms per months who were obese as per the World Health Organisation (WHO) definition (Table 1). This finding persisted in regression analysis adjusted for age, sex, underlying heart disease, indication for WCD and total number of days of WCD use. The mean BMI of patients without any alarms was 25.7 kg/m^2^ (±5.3), compared to 27 kg/m^2^ (±6.1) in patients with at least one alarm. Association of a higher alarm burden due to analysed artefact recordings is shown in Appendix A.

Significantly more patients with a clinical diagnosis of AF had at least one episode of recorded AF, as expected (63.6% vs. 22.7%, *p* = 0.002). On the other hand, no patient with a clinical diagnosis of other supraventricular tachycardia had an episode of supraventricular tachycardia recorded by the WCD.

### 3.3. Impact of Alarms on Stopping WCD Use and ICD Implantation

WCD use was discontinued most commonly due to ICD implantation (212, 47.6%) (Table 2). In the remaining patients, the majority (183, 75.6%) had no further indication for ICD. Comfort issues or patient choice were reasons for stopping WCD use in 45 patients (9.9%). Of note, ≥1 alarm per day did not lead to a significantly higher rate of stopping the WCD for comfort reasons (Table 3).

A higher rate of WCD alarms per month, did not have a significant impact on ICD implantation rates. In fact, the ICD implantation rate was marginally higher in patients with three or more alarms per month (Table 2). On the other hand, patients with at least one episode of recorded VT/VF had a significantly higher ICD implantation rate after stopping WCD use (85.7% vs. 45.8%, *p* < 0.001). These patients also had a trend towards a higher rate of appropriate ICD therapies (22.2% vs. 9.2%, *p* = 0.081) during follow-up (Appendix A).

### 3.4. Patients with Very High Alarm Burden

Forty-nine patients (10.7%) had at least one alarm per day on average emitted by their WCD over the complete wear period. There was a trend towards a higher high alarm burden in men (*p* = 0.063), whereas other baseline characteristics were comparable between patients with 1 or more alarms per day and those with less frequent alarms. Importantly, the very high number of alarms had neither a significant impact on the reason for stopping the use of the WCD nor on the rate of ICD implantation in these patients (Table 3).

## 4. Discussion

Several clinically relevant findings deserve to be highlighted from the analysis of our nationwide registry: (1) Almost half of all patients with WCD had at least three alarms emitted per month by their device; (2) a very high alarm burden (at least one alarm per day) was noted in approximately 11% of patients over the complete WCD wear period, most of these being male patients; (3) the higher alarm burden, however, did not lead to a decreased adherence, as determined by average daily wear-times; (4) obesity was significantly associated with a higher alarm burden; and (5) most alarms were due to artefact or rapidly conducted atrial fibrillation.

Alarms have an important role in the mechanism of WCD. Due to limited reliability of surface ECG recordings, a second fail-safe is available. If a VT/VF is detected by the WCD, a series of vibratory and auditory alarms are emitted by the device, and patients can manually abort the delivery of DC, if they are still conscious. In case of ongoing arrhythmia detection, the alarms are deployed repeatedly, which may possibly lead to anxiety and a decreased quality-of-life in these patients.

It has been demonstrated that inappropriate ICD shocks can negatively impact prognosis of patients [3,4]. Therefore, it is also of paramount importance for patients using the WCD, especially in the susceptible early phase of their disease process, to prevent inappropriate delivery of shocks. Here, for the first time, we describe the impact of impending shock alarm burden on clinical outcomes from the database of a large national registry. Other studies have reported common occurrence of inappropriate detection and device alarms, mostly due to artefacts [13,14,15]. In our cohort, we also report a high number of alarms, most of which were caused by short detected arrhythmic episodes and detection artefacts. It is obvious that such alarms can have a deleterious effect on the quality-of-life of patients and may even impact further adherence to prescribed therapy.

To date, only small retrospective reports on the quality-of-life in patients with WCD use have been published. A study on 123 patients who were eligible for a WCD, depression and anxiety symptoms were common (21% and 52%, respectively), and WCD recipients showed similar changes of depression and anxiety at 6 weeks when compared to non-recipients [16]. Another small study reported a high rate of sleep disturbances (48%), fear of shock (29%), but also feeling of safety (64%) [13]. The authors noted a significantly higher rate of fear of shock in patients with frequent alarms. In our study, the higher alarm burden did not negatively impact the average daily wear-time, the decision to stop WCD use or medication used. Importantly, ICD implantation rate after stopping WCD use was also not significantly affected by higher alarm burden. Although we did not evaluate the impact of alarm burden on objective measures of quality-of-life, we assessed if stopping WCD use was due to discomfort or patient choice. Interestingly, a similar percentage of patients with higher and lower alarm burden chose to terminate WCD use. This suggests that, although a significant number of patients stop WCD use with persisting indication, the alarm burden does not seem to significantly alter this decision. The reason for the lack of detrimental effect of alarm burden on termination of WCD use is unclear. We speculate that the more frequent consultations and care by the ZOLL support team and as a consequence the treating physician may increase adherence in general, including the use of WCD. This has been suggested to have caused the decrease in mortality in the VEST trial despite the lack of significant effect on arrhythmic death [11].

The high rate of inappropriate arrhythmia detection was directly responsible for the high alarm burden in our study. The efficacy of WCD shock delivery and arrhythmia conversion rate in demographics based on age and BMI has already been established [17,18]. However, the varying rate of alarm burden has not been investigated in this context. We found that obese patients were significantly more likely to have a higher alarm burden. A decreased skin-contact of the ECG electrodes and movement of subcutaneous tissue due to the patients’ body constitution may explain this finding. Wan et al. investigated the efficacy of WCD shocks in obese patients. They found a similar conversion rate irrespective of BMI, although the impedance measured by the defibrillator pads directly correlated to the BMI. Of note, an increased body weight did not increase the rate of inappropriate shocks. However, the higher alarm burden, which we have demonstrated in obese patients in our study, suggests that these patients more frequently need to be alert and manually abort an impending inappropriate shock.

In our study, of all VT/VFs, 80% were sustained episodes, for which patients manually inhibited shock administration, whereas shock delivery occurred in only a small percentage of the patients. The higher rate of sustained VT/VF episodes compared to non-sustained episodes was possibly due to the shorter episodes not being registered by the WCD by default. Although ICD implantation was significantly more frequent in patients with at least one episode of documented VT/VF, 14% of these patients did not later undergo ICD implantation, which was largely due to patients’ choice and terminal patient status. According to current guidelines, ICD implantation is indicated in the presence of hemodynamically relevant VT [19,20]. Data on the prognostic impact of ICD implantation in patients with hemodynamically tolerated VT, such as patients capable of inhibiting of shock administration by the WCD, are lacking from randomized-controlled trials. On the other hand, sustained or non-sustained VT episodes may have prognostic implications in patients with heart failure [4,21]. Indeed, we observed a trend in patients with at least one VT/VF episode recorded by the WCD and subsequent ICD implantation after stopping WCD use, towards a higher prevalence of appropriate ICD therapies shorter time to therapy.

Attention needs to be given to patients with high alarm burdens, and not only to patients with appropriate alarms due to VT/VF episodes. Particular attention is needed for obese patients. Even though we did not detect an increased rate of inappropriate shocks or a decrease in adherence to therapy, the likely psychological effect on patients should not be underestimated. Alarms during the night-time and longer alarms could theoretically induce more stress in WCD patients. Unfortunately, data on the time and duration of individual alarms were not available in our registry. Inquiring the frequency of WCD alarms should be part of routine clinical follow-up and effort should be made to minimize alarm burden due to artefacts (i.e., by refitting the vest of the WCD). Moreover, examining activity levels of patients and correlating them to the alarm burden is a recently emerging possibility by the TRENDS function of the WCD [22,23].

Our study has several limitations. Data on the impact of alarm burden on validated quality of life scores would be of interest but were not available. Furthermore, arrhythmia recordings less than 30 s in length were not systematically analysed.

## 5. Conclusions

In patients using WCDs, alarms emitted by the device in the form of vibratory and auditory alerts and impending inappropriate shocks were frequent and were most commonly caused by artefacts. A high alarm burden did not lead to a decreased adherence, as determined by average daily wear-times. Obesity was significantly associated with a higher alarm burden.

## Figures and Tables

**Figure 1 jcm-10-03811-f001:**
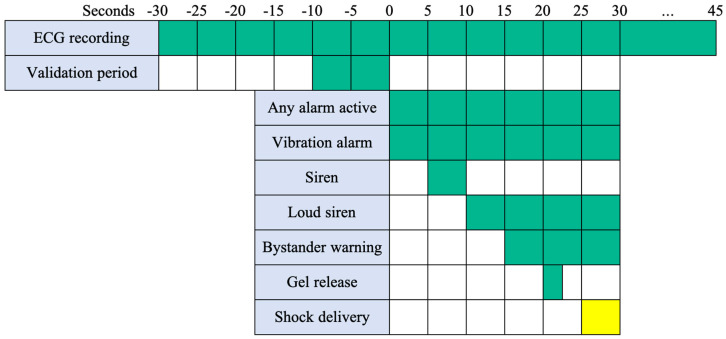
Alarms and treatment sequences. The timing of alarm sequences and delivery of shock depicted in five seconds steps. Figure is adapted from ZOLL.

**Figure 2 jcm-10-03811-f002:**
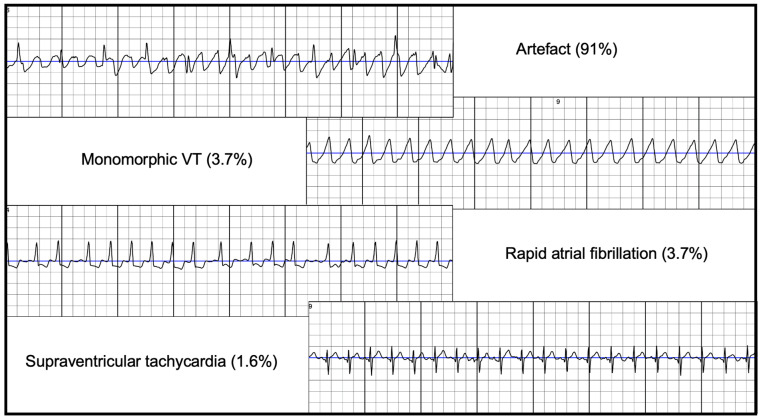
Categories of analysed recordings (*n* = 2996), VT = ventricular tachycardia.

**Table 1 jcm-10-03811-t001:** Baseline characteristics of patients in the Swiss WCD Registry.

	<3 Alarms Per Month (*n* = 249)	≥3 Alarms Per Month (*n* = 207)	Total (*n* = 456)	*p* Value
**Age**	57.9 (14.0)	56.6 (14.0)	57.3 (14.0)	0.325
**Sex**				0.663
Female	46 (18.5%)	35 (16.9%)	81 (17.8%)	
Male	203 (81.5%)	172 (83.1%)	375 (82.2%)	
**BMI (kg/m^2^)**				0.01 ^#^
<18	5 (2.0%)	4 (2.0%)	9 (2.0%)	
18–29	201 (82.0%)	142 (70.3%)	343 (76.7%)	
≥30	39 (15.9%)	56 (27.7%)	95 (21.3%)	
**LVEF**	30.7 (13.3)	32.5 (12.9)	31.5 (13.1)	0.140
**Indication for WCD**				0.113
Low LVEF	96 (38.6%)	56 (27.1%)	152 (33.3%)	
Recent myocardial infarction	62 (24.9%)	63 (30.4%)	125 (27.4%)	
Bridging *	30 (12.0%)	39 (18.8%)	69 (15.1%)	
Recent PCI or CABG with low LVEF	23 (9.2%)	13 (6.3%)	36 (7.9%)	
VT with normal LVEF	18 (7.2%)	17 (8.2%)	35 (7.7%)	
Percutaneous or surgical valve repair with low EF	8 (3.2%)	5 (2.4%)	13 (2.9%)	
Syncope	3 (1.2%)	3 (1.4%)	6 (1.3%)	
Other	9 (3.6%)	11 (5.3%)	20 (4.4%)	
**Underlying heart disease**				0.358
Ischemic cardiomyopathy without coronary dissection	140 (56.2%)	133 (64.3%)	273 (59.9%)	
Non-ischemic cardiomyopathy	65 (26.1%)	43 (20.8%)	108 (23.7%)	
Valvular	15 (6.0%)	8 (3.9%)	23 (5.0%)	
Myocarditis	5 (2.0%)	5 (2.4%)	10 (2.2%)	
Channelopathy	7 (2.8%)	2 (1.0%)	9 (2.0%)	
Congenital	2 (0.8%)	5 (2.4%)	7 (1.5%)	
Ischemic cardiomyopathy with coronary dissection	1 (0.4%)	1 (0.5%)	2 (0.4%)	
Other	14 (5.6%)	10 (4.8%)	24 (5.3%)	
**Medical therapy**				
Betablocker	227 (91.2%)	187 (90.3%)	414 (90.8%)	0.761
ACEi, ARB or neprilysin inhibitor	219 (88.0%)	181 (87.4%)	400 (87.7%)	0.868
Aldosterone antagonist	147 (59.0%)	132 (63.8%)	279 (61.2%)	0.302
Amiodarone	43 (17.3%)	37 (17.9%)	80 (17.5%)	0.866
**Atrial fibrillation**	59 (23.7%)	49 (23.7%)	108 (23.7%)	0.995
**Atrial flutter**	19 (7.6%)	16 (7.7%)	35 (7.7%)	0.968
**Other supraventricular tachycardia**	8 (3.2%)	3 (1.4%)	11 (2.4%)	0.222

Data are displayed according to number of alarms emitted by the WCD per month. Mean (SD) and number (%). Statistical analysis was performed with chi square, fisher’s exact test or univariate regression analysis, as appropriate. * Bridging until ICD-reimplantation, until primary ICD implantation or until heart transplant. ^#^ Statistically significant finding. Abbreviations: ACEi = angiotensin converting enzyme inhibitor; ARB = angiotensin II receptor antagonist; BMI = body mass index; CABG = coronary artery bypass graft; LVEF = left ventricular ejection fraction; PCI = percutaneous coronary intervention; VT = ventricular tachycardia.

**Table 2 jcm-10-03811-t002:** WCD wear data and outcomes.

	<3 Alarms Per Month (*n* = 249)	≥3 Alarms Per Month (*n* = 207)	Total (*n* = 456)	*p* Value
**Average wear hours per day**	20.7 (4.4)	20.8 (3.8)	20.8 (4.2)	0.785
**Reason for stopping WCD use**				0.610
ICD implantation	106 (42.6%)	90 (43.5%)	196 (43.0%)	
Normalized arrhythmic risk	91 (36.5%)	76 (36.7%)	167 (36.6%)	
Comfort issue or patient choice	22 (8.8%)	23 (11.1%)	45 (9.9%)	
Unknown	30 (12.0%)	18 (8.7%)	48 (10.5%)	
**LVEF after WCD use**	38.7 (12.1)	38.0 (13.3)	38.4 (12.6)	0.592
**Device implanted**				0.156
ICD	115 (47.3%)	97 (48.0%)	212 (47.6%)	
PM	0 (0.0%)	3 (1.5%)	3 (0.7%)	
None	128 (52.7%)	102 (50.5%)	230 (51.7%)	
**Reason for not implanting an ICD**				0.454
Not indicated	101 (75.4%)	82 (75.9%)	183 (75.6%)	
Patient choice	10 (7.5%)	10 (9.3%)	20 (8.3%)	
Terminal	2 (1.5%)	4 (3.7%)	6 (2.5%)	
Other	2 (1.5%)	3 (2.8%)	5 (2.1%)	
Unknown	19 (14.2%)	9 (8.3%)	28 (11.6%)	
**Follow-up duration (days) ***	545.4 (383.6)	478.6 (383.5)	514.3 (384.1)	0.225
**First treatment by ICD after implantation**				0.471
None	98 (86.0%)	89 (89.0%)	187 (87.4%)	
Adequate treatment	12 (10.5%)	10 (10.0%)	22 (10.3%)	
Inadequate treatment	4 (3.5%)	1 (1.0%)	5 (2.3%)	
**Time to treatment after ICD implantation (days)**	232.5 (287.9)	357.8 (328.2)	286.2 (306.4)	0.293

Data are displayed mean (SD) and as number (%). Statistical analysis was performed with chi square, fisher’s exact test or univariate regression analysis, as appropriate. * follow-up duration for patients implanted with an ICD, includes the duration of WCD use and follow-ups in the device clinic. Abbreviations: ICD = implantable cardioverter-defibrillator; LVEF = left ventricular ejection fraction; PM = pacemaker, WCD = wearable cardioverter-defibrillator.

**Table 3 jcm-10-03811-t003:** Alarm burden in WCD patients.

	<1 Alarm Per Day (*n* = 407)	≥1 Alarm Per Day (*n* = 49)	Total (*n* = 456)	*p* Value
**Average wear hours/day**	20.8 (4.0)	20.0 (5.0)	20.8 (4.2)	0.161
**Reason for stopping WCD use**				0.521
Normalized arrhythmic risk	147 (36.1%)	20 (40.8%)	167 (36.6%)	
ICD implantation	173 (42.5%)	23 (46.9%)	196 (43.0%)	
Comfort issue or patient choice	42 (10.3%)	3 (6.1%)	45 (9.9%)	
Unknown	45 (11.1%)	3 (6.1%)	48 (10.5%)	
**Reason for not implanting an ICD**				0.007 ^#^
Not indicated	163 (74.8%)	20 (83.3%)	183 (75.6%)	
Patient choice	20 (9.2%)	0 (0.0%)	20 (8.3%)	
Unknown	28 (12.8%)	0 (0.0%)	28 (11.6%)	
Terminal stadium ^$^	4 (1.8%)	2 (8.3%)	6 (2.5%)	
Other	3 (1.4%)	2 (8.3%)	5 (2.1%)	

Data are displayed mean (SD) and as number (%). Statistical analysis was performed with chi square, fisher’s exact test or univariate regression analysis, as appropriate. Abbreviations: ICD = implantable cardioverter-defibrillator; WCD = wearable cardioverter-defibrillator. ^#^ statistically significant results; ^$^ terminally ill patients, change of management to comfort care.

## Data Availability

Data used in this study can be made available upon reasonable request.

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
