# Peer review of "High Incidence of Inappropriate Alarms in Patients with Wearable Cardioverter-Defibrillators: Findings from the Swiss WCD Registry"

_jcm, 2021, doi:10.3390/jcm10173811_

Round 1
Reviewer 1 Report
I have no important comments to the paper. The problem is of great significant importance. The results describes the real world results with the wearable cardioverter defibrillators that is a relatively new and developing technology.
Author Response
Dear reviewer
We appreciate your positive feedback. We have not performed any changes accordingly.
Reviewer 2 Report
Dear Authors,
I recently had the pleasure of reviewing your manuscript entitled "High Incidence of Inappropriate Alarms in Patients with Wear-able Cardioverter-Defibrillators: Findings from the Swiss WCD Registry".
This manuscript addresses a very important, yet under-addressed, topic, of clear importance for the field of cardiology and device-management.
The real-world sample size used for this multi-centered analysis is significant (n=456 patients, with a median of 2 months of wearing) and captures the complexity and inhomogeneity of a clinical problem of great relevance for the every day practice.
Data have been presented in an appropriate fashion. Conclusions are reasonable supported by the results of the study. Authors should be commended for their effort.
I only have minor comments:
- I'd suggest removing the "Associations with recorded VT/VF episodes" paragraph from the results section, being this very brief. The data is easily accessible from the clear tables that have been provided.
- BMI is often described as a potential problem for most detection algorithms in wearables. Unsurprisingly, authors report a difference in BMI class distributions among the groups in Table1. I think this is a key point in their manuscript, which is correctly highlighted in the discussion.
If BMI is available as a continuous variable from the dataset of the registry, I'd suggest using a logistical/linear regression analysis, to test if there is a correlation between BMI and a) the presence of alarms at all; b) the overall number of alarms. I think this could be useful information for the readers. - A personal pet-peeve: in the statistical analysis section and in the descriptive tables, authors mention the use of univariate regression analysis. I wonder if there is a particular reason behind this, compared to the direct use of a t-test.
Thank you for this great learning opportunity
Author Response
Dear reviewer
We appreciate your constructive feedback and have addresses the three comments:
- We agree that the section "Associations with recorded VT/VF episodes" can be viewed by readers in the tables, therefore we have removed it.
- We also believe that a regression/linear analysis of BMI and number of alarms is of importance due to the continuous nature of the variable. We have indeed performed a regression analysis showing a highly significant correlation (<0.001) of number of alarms with increasing BMI (95% CI 1.01-1.02) with however a very flat curve in the plot analysis. This was deemed after a prolongued discussion with our co-authors due to the fact that 28.9% of patients had 0 alarms and alarm distribution was skewed in the remaining patients. Therefore we have decided not to primarily report it to prevent overemphasis of the highly significant p-value in this case and continue with the categorised variable. We have however added for the reviewer's benefit this information to the supplementary materials.
- We thank for this insightful comment from the reviewer. We have decided for the use of a regression analysis in case a multivariate analysis on the same dataset was necessary. Since the findings, in particular, regarding the BMI were highly significant, we do not expect any relevant effect on the results if additional analyses were made with t test.